# SCALE-DIT: ULTRA-HIGH-RESOLUTION IMAGE GENERATION WITH HIERARCHICAL LOCAL ATTENTION

**Yuyao ZHANG**
Department of Computer Science
Dartmouth College

**Yu-Wing Tai**
Department of Computer Science
Dartmouth College

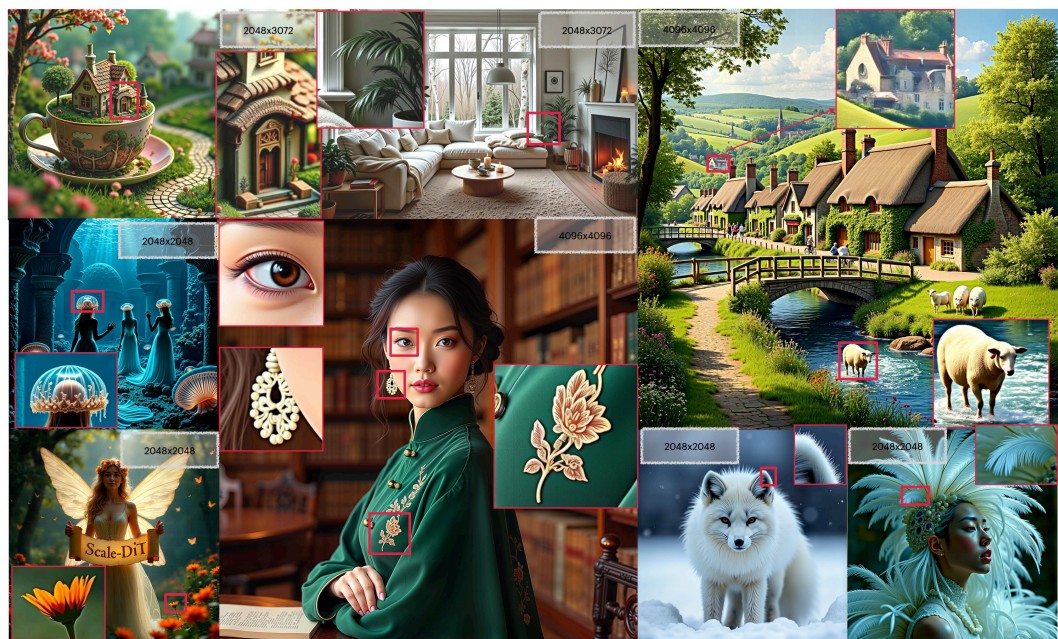

Figure 1: Ultra-high-resolution images generated by **Scale-DiT** at $4K \times 4K$, $2K \times 3K$, and $2K \times 2K$. Zoomed-in regions highlight the fine-grained details preserved at these scales.

## ABSTRACT

Ultra-high-resolution text-to-image generation demands both fine-grained texture synthesis and globally coherent structure, yet current diffusion models remain constrained to sub-$1K \times 1K$ resolutions due to the prohibitive quadratic complexity of attention and the scarcity of native $4K$ training data. We present **Scale-DiT**, a new diffusion framework that introduces hierarchical local attention with low-resolution global guidance, enabling efficient, scalable, and semantically coherent image synthesis at ultra-high resolutions. Specifically, high-resolution latents are divided into fixed-size local windows to reduce attention complexity from quadratic to near-linear, while a low-resolution latent equipped with scaled positional anchors injects global semantics. A lightweight LoRA adaptation bridges global and local pathways during denoising, ensuring consistency across structure and detail. To maximize inference efficiency, we repermute token sequence in Hilbert curve order and implement a fused-kernel for skipping masked operations, resulting in a GPU-friendly design. Extensive experiments demonstrate that **Scale-DiT** achieves more than $2\times$ faster inference and lower memory usage compared to dense attention baselines, while reliably scaling to $4K \times 4K$ resolution without requiring additional high-resolution training data. On both quantitative benchmarks (FID, IS, CLIP Score) and qualitative comparisons, **Scale-DiT** delivers superior global coherence and sharper local detail, matching or outperforming state-of-the-art methods that rely on native 4K training. Taken together, these results highlight hierarchical local attention with guided low-resolution anchors as a promising and effective approach for advancing ultra-high-resolution image generation.

# 1 INTRODUCTION

Ultra-high-resolution image generation is becoming increasingly crucial for both creative and practical applications. From digital art and advertising to scientific visualization and virtual environments, users demand outputs with fine-grained textures, faithful structures, and seamless details at $4K$ resolution or higher. However, despite the rapid progress of text-to-image (T2I) diffusion models Ding et al. (2021); Rombach et al. (2022); Chen et al. (2023); Podell et al. (2023); Peebles & Xie (2023); Xue et al. (2023); Li et al. (2024b); Chen et al. (2024b); Li et al. (2024a); Black Forest Labs (2025); Cai et al. (2025); Xie et al. (2024; 2025), most state-of-the-art systems remain confined to resolutions below $1K \times 1K$. This limitation fundamentally restricts their usability in scenarios where ultra-high fidelity is paramount.

The root of this challenge lies in two interdependent bottlenecks. First, scaling model capacity and datasets to higher resolutions requires tremendous resources, as high-quality $2K$–$8K$ training data are scarce and expensive to curate. Second, attention-based diffusion architectures suffer from quadratic growth in token complexity with respect to image resolution, making naive scaling computationally prohibitive. For example, generating a $4K$ image would require tens of thousands of tokens, leading to impractical memory and runtime costs. As a result, existing approaches either resort to costly retraining on synthetic high-resolution datasets Hoogeboom et al. (2023); Liu et al. (2024a); Ren et al. (2024); Teng et al. (2023); Zheng et al. (2024); Zhang et al. (2025b), or adopt training-free upscaling methods Guo et al. (2024); Qiu et al. (2024); Du et al. (2024b); Liu et al. (2024b); Wu et al. (2025); Shi et al. (2025); Kim et al. (2025); Huang et al. (2024); Bu et al. (2025) that often compromise efficiency and stability. Consequently, the field still lacks a solution that delivers *both* ultra-high-resolution fidelity and computational scalability.

In this work, we introduce **Scale-DiT**, a novel framework that rethinks ultra-high-resolution synthesis through the lens of efficiency and scalability. Inspired by the way artists construct large-scale murals, we decompose the generation task hierarchically: local windows capture fine textures through lightweight attention, while a low-resolution guidance image preserves global semantic structure. Technically, **Scale-DiT** introduces three innovations. First, a *hierarchical local attention* mechanism restricts computation to fixed-size windows, reducing quadratic cost to near-linear scaling while yielding over $10GB$ memory savings and more than $2\times$ faster inference compared to dense attention. Second, a *global guidance pathway* employs a low-resolution latent with scaled RoPE positional anchors to maintain long-range dependencies and ensure semantic coherence across windows. Third, a *parameter-efficient joint denoising framework* integrates global and local pathways via LoRA-adapted projections trained only on $256 \times 256$–$1K \times 1K$ data, enabling direct scaling to $4K$ or higher without requiring any native high-resolution training. These design choices translate into two key outcomes: (1) *scalable fidelity*, achieved by hierarchically fusing global composition with local detail to extend beyond pretrained resolutions, and (2) *computational efficiency*, enabled by reducing computation, memory, and runtime while reusing pretrained weights with lightweight adaptation.

Extensive experiments validate these contributions: **Scale-DiT** reliably scales to $4K \times 4K$ synthesis with commodity training resolutions, achieving sharper textures, richer details, and stronger global coherence than existing approaches. Quantitatively, it surpasses dense attention baselines with over $2\times$ faster inference and lower memory usage, and it matches or outperforms state-of-the-art methods trained with native 4K data across FID, IS, and CLIP Score. Together, these results highlight **Scale-DiT** as a practical and effective solution for advancing ultra-high-resolution text-to-image generation.

We summarize our contributions as follows:

- We propose a new paradigm for ultra-high-resolution text-to-image generation that combines local window attention with low-resolution global guidance, enabling scalable synthesis beyond pretraining limits.

- Our framework entirely eliminates the need for high-resolution training data, instead leveraging positionally aligned low-resolution guidance to preserve global coherence.

- We demonstrate substantial efficiency improvements ($> 2\times$ speedup, less memory usage) while achieving state-of-the-art quality at $4K$, offering a practical and generalizable solution for real-world ultra-high-resolution generation.

## 2 RELATED WORK

**Text-to-Image Generation.** Recent years have witnessed rapid progress in text-to-image (T2I) generation Ding et al. (2021); Rombach et al. (2022); Chen et al. (2023); Podell et al. (2023); Peebles & Xie (2023); Xue et al. (2023); Li et al. (2024b); Chen et al. (2024b); Li et al. (2024a); Black Forest Labs (2025); Cai et al. (2025); Xie et al. (2024; 2025); Gao et al. (2025), where diffusion models now achieve near-photorealistic synthesis at resolutions up to $1K \times 1K$. The dominant paradigm relies on cross-attention mechanisms and DiT architecture Peebles & Xie (2023), as seen in the PixArt series Chen et al. (2023; 2024b;a), or multi-modal diffusion transformers (MMDiT) such as FLUX.1.0-dev Black Forest Labs (2025). These architectures have established strong foundations for controllable and high-quality T2I synthesis. However, despite their success, existing methods struggle to scale beyond pretrained resolutions due to quadratic attention costs and the lack of large-scale high-resolution data. Our work builds directly on this line of research, but departs from the prevailing direction by focusing on resolution scalability and computational efficiency rather than retraining larger models.

**High-resolution image synthesis.** Real-world applications increasingly demand resolutions of $4K$ or higher, sparking intense research interest in pushing beyond the $1K \times 1K$ barrier. Several works, including PixArt-$\Sigma$ and SANA 1.5 Xie et al. (2025), have achieved near-$4K$ synthesis through extensive high-resolution pretraining, while others Hoogeboom et al. (2023); Liu et al. (2024a); Ren et al. (2024); Xie et al. (2023); Teng et al. (2023); Zheng et al. (2024); Zhang et al. (2025b); Yu et al. (2025) pursue fine-tuning or training-from-scratch approaches using curated datasets. For example, Zhang et al. (2025b) used wavelet supervision to enhance detail clarity, and Yu et al. (2025) proposed lightweight fine-tuning for adapting to higher resolutions. Although effective, these methods remain constrained by the scarcity of high-quality high-resolution data and substantial GPU requirements. More recent training-free strategies Guo et al. (2024); Qiu et al. (2024); Du et al. (2024b); Liu et al. (2024b); Wu et al. (2025); Shi et al. (2025); Kim et al. (2025); Huang et al. (2024); Bu et al. (2025); Zhang et al. (2025g); ZHANG et al. avoid data collection by leveraging pretrained models directly. For instance, I-Max Du et al. (2024b) aligns high-resolution flow with low-resolution manifolds, and HiFlow Bu et al. (2025) introduces low-resolution initialization to guide denoising. While promising, these approaches often inherit significant runtime and memory overhead, limiting accessibility for broader use. Other recent methods, such as HiDiffusion Zhang et al. (2025g) and FreCaS ZHANG et al., both target UNet-based architectures. HiDiffusion is training-free and regularizes the denoising trajectory but still struggles to preserve long-range consistency, while FreCaS employs a coarse-to-fine strategy to synthesize only high-frequency content, which reduces the long-range dependency burden but requires careful stage alignment and CFG tuning. In contrast, our method explicitly decouples fine-detail generation from global semantic reasoning using local-window attention and low-resolution guidance, achieving stable and efficient ultra-high-resolution synthesis without multi-stage repair or delicate parameter balancing.

**Attention Acceleration.** As image and video resolutions increase, the quadratic complexity of attention becomes the dominant bottleneck. Works including the SANA series Xie et al. (2024; 2025); Zhu et al. (2025) leverage linear attention to reduce complexity. While such methods achieve satisfactory performance, the non-injective property and loss of attention spikiness Han et al. (2024); Meng et al. (2025); Zhang et al. inherent in linear attention lead to confusion and inconsistent local details in real-world scenarios. For softmax attention, system-level optimizations such as Flash Attention Dao et al. (2022); Dao (2023); Shah et al. (2024) exploit GPU features for faster execution, while quantization Zhang et al. (2025d;c) and sparsity-based designs Deng et al. (2024); Liu et al. (2022) reduce computational load. Architectural innovations, such as LongFormer Beltagy et al. (2020) and SwinFormer Liu et al. (2021), employ local attention patterns, and more recent works Lai et al. (2025); Zhang et al. (2025e); Xu et al. (2025); Xi et al. (2025); Yang et al. (2025); Yuan et al. (2024); Zhang et al. (2025a;f); Liu et al. (2025); Ren et al. (2025) propose block sparsification or compression strategies for diffusion transformers. Although these techniques yield noticeable acceleration, the gains remain insufficient for ultra-high-resolution synthesis, and compression often risks degrading fine-grained fidelity. Methods like Liu et al. (2025); Ren et al. (2025); Zhang et al. (2025f) aim to achieve faster generation while preserving local image quality using local neighbor attention. However, none of them fully maintain long-range dependencies: CLEAR Liu et al. (2025) and GRAT Ren et al. (2025) require increasingly large local windows/grouping as resolution increases, while STA Zhang et al. (2025f) sacrifices some long-range consistency. In contrast, our method explicitly maintains long-range consistency at low resolution through hierarchical local-window at-

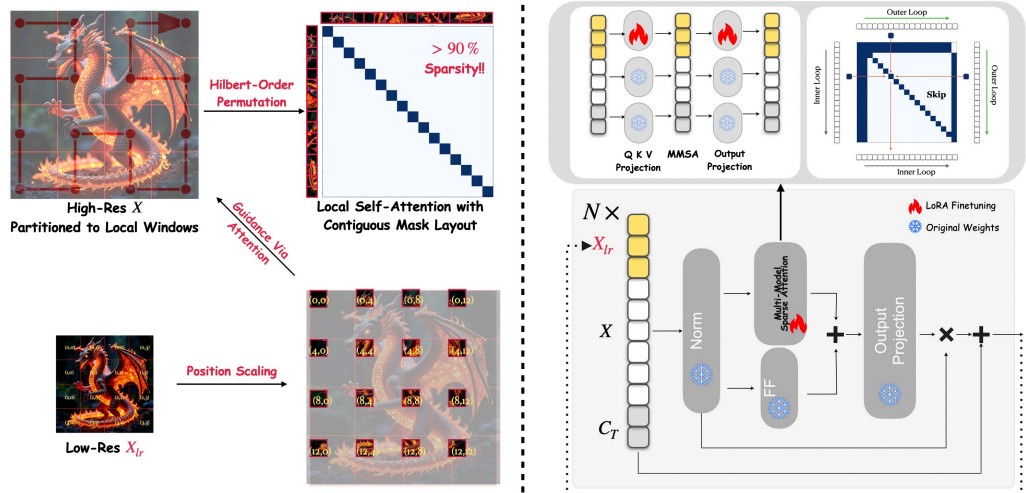

Figure 2: Schematic of **Scale-DiT**'s attention block modifications. The left column illustrates that high-resolution image latents $X$ are partitioned into local windows (red) in Hilbert-Curve order that attend to each other within their window. Simultaneously, a low-resolution guidance latent $X_{lr}$ (yellow) provides global context to each window via position scaling. The right column shows the joint-denoising process and the attention kernel, that $X$, $X_{lr}$ are processed together with LoRA applied on the $X_{lr}$ part. The attention mask (upper right) enforces local and guidance-specific interactions and tile skipping in attention calculation to enable efficient and coherent ultra-high-resolution generation. No additional high-resolution training data are needed.

tention, achieving stronger acceleration ($> 2\times$) and better detail preservation when scaling to 4K, without relying on excessively large local windows or compromising global structure.

## 3 METHOD

Figure 2 illustrates the overall framework of **Scale-DiT**, highlighting the key modifications introduced in the attention blocks to support ultra-high-resolution generation. High-resolution image latents $X$ are partitioned into local windows (red grids), where tokens attend only within their window to capture fine-grained detail efficiently. A low-resolution guidance latent $X_{lr}$ (yellow), enhanced with scaled positional anchors, injects global semantics and preserves long-range consistency across windows. The attention mask is designed to enforce both local and guidance-specific interactions while also enabling tile-skipping to avoid unnecessary computation. Together, these mechanisms allow **Scale-DiT** to generate coherent and detail-rich $4K$ images with near-linear complexity and without requiring any native high-resolution training data. This schematic provides the foundation for the following technical components.

**The Multi-Modal Diffusion Transformer (MMDiT) Preliminaries**. The MMDiT employed in state-of-the-art models Black Forest Labs (2025); Cai et al. (2025); Gao et al. (2025), represents the current benchmark architecture for text-to-image generation. MMDiT processes two distinct token modalities: a noisy image token sequence $X \in \mathbb{R}^{N \times d}$ and a text token sequence $C_T \in \mathbb{R}^{M \times d}$. MMDiT processes image and text tokens through a unified Multi-Modal Attention (MMA) mechanism. Specifically, image and text tokens are concatenated as $[C_T; X]$ and processed via self-attention.

Within the MMA framework, spatial information is encoded using Rotary Position Embedding (RoPE) Su et al. (2024), which applies rotational transformations to capture relative positional relationships. This is mathematically represented as: $X_{i,j} = X_{i,j} \cdot \text{Rot}(i, j)$, where $\text{Rot}(i, j)$ denotes the rotation matrix corresponding to position $(i, j)$ in the 2D spatial grid. The MMA mechanism is formally defined as:

$$\text{MMA}([C_T; X]) = \text{Softmax}\left(\frac{([Q_T(C_T), Q(X)])\,([K_T(C_T), K(X)]^T)}{\sqrt{d}}\right)[V_T(C_T), V(X)]$$

where $Q, Q_T, K, K_T$, and $V, V_T$ represent the query, key, and value projection matrices applied to the position-encoded image and text tokens $X, C_T$. This formulation enables bidirectional attention across all token modalities.

**Efficient Local Window Attention.** Transformer-based generative models rely on self-attention, whose computational and memory costs scale quadratically with the number of tokens. For an image of size $H \times W$, the number of spatial tokens is $N = H \cdot W$, leading to $O(N^2)$ complexity. Scaling from $1024 \times 1024$ (4096 tokens[1]) to $4096 \times 4096$ (65536 tokens) increases the cost by a factor of 256, which is prohibitive even with optimized kernels such as FlashAttention Dao et al. (2022); Dao (2023); Shah et al. (2024).

To overcome this limitation, we partition the high-resolution latent $X \in \mathbb{R}^{H \times W}$ into non-overlapping windows $x_i \in \mathbb{R}^{l \times l}$, with window size $l$ bounded by the pretrained resolution (e.g., 1024). Self-attention is computed independently within each window, reducing the complexity from $O(N^2)$ to $O\left(\lceil \frac{H}{l} \rceil \cdot \lceil \frac{W}{l} \rceil \cdot (l^2)^2\right) \approx O(N \cdot l^2)$. In practice, we set $l = 16$ (corresponding to $256 \times 256$ windows) to strike a balance between accuracy and efficiency: larger windows bring little additional benefit in quality, while smaller windows underutilize GPU kernels, which are optimized for tile sizes around $128 \times 128$[2]. An ablation study in Section 5 further analyzes this trade-off.

With this design, scaling to $4K$ resolution ($N = 65536$ tokens) reduces the attention complexity from $O(65536^2)$ to $O(65536 \cdot 16^2)$, yielding a $256\times$ reduction. To maintain boundary consistency, we also allow limited cross-window attention along adjacent regions. Crucially, by fixing the window size regardless of the overall resolution, the local window cost remains constant as $H$ and $W$ grow. This makes the runtime scale nearly linearly with image resolution, enabling efficient and memory-friendly ultra-high-resolution generation while preserving fine-grained detail within each window.

**Maintaining Global Semantics via Low-Resolution Guidance.** Partitioning a high-resolution image into local windows risks creating discontinuities and losing global semantic coherence. The key insight is that global long-range dependencies primarily influence the overall structure and layout of the generated image, but have minimal effect on fine local details. We generate a low-resolution guidance image $X_{lr} \in \mathbb{R}^{h \times w}$ to provide global context. The corresponding position indices are denoted as $(m, n)$, where $m \in \{0, 1, \ldots, h-1\}$ and $n \in \{0, 1, \ldots, w-1\}$. We define the scaling ratio as $\rho = \frac{H}{h}$ (empirically set to 4 for an optimal balance of performance and efficiency). We then scale the low-resolution image's position indices by this ratio, mapping them to $(\tilde{m}, \tilde{n}) = (\rho \cdot m, \rho \cdot n)$. This effectively projects $X_{lr}$ to the same spatial scale as the high-resolution image $X$. Each token $X_{lr}[\tilde{m}, \tilde{n}]$ acts as an anchor point, providing contextual information. Each high-resolution window attends to its local tokens and the corresponding scaled region in $X_{lr}$, while $X_{lr}$ tokens attend globally among themselves and to text tokens. This ensures semantic consistency across distant regions. The framework naturally supports recursive generation: the high-resolution output can serve as low-resolution guidance for an even higher resolution, enabling arbitrarily large-scale synthesis with stable memory and computation.

**Parameter-Efficient Joint Denoising.** To integrate the low-resolution guidance $X_{lr}$, we concatenate it with the standard text–image sequence to form $[C_T, X, X_{lr}]$. Since $X_{lr}$ is still an image modality, we can reuse the pretrained VAE and DiT blocks for its processing. However, scaling the positional indices of $X_{lr}$ alters the frequency characteristics of the RoPE embeddings. To adapt to this change, we fine-tune only the query, key, and value projections $(Q, K, V)$ that process $X_{lr}$ using LoRA Hu et al. (2022), yielding adapted matrices $\tilde{Q}, \tilde{K}, \tilde{V}$. Importantly, the original $Q, K, V$ are still used for the high-resolution windows $X$, ensuring that pretrained generative capabilities for local content are preserved. The unified attention mechanism, simplified by omitting text tokens, is then:

$$\text{MMA}([X; X_{lr}]) = \text{Softmax}\left(\frac{\left([Q(X), \tilde{Q}(X_{lr})]\right)([K(X), \tilde{K}(X_{lr})]^T) \cdot M}{\sqrt{d}}\right)[V(X), \tilde{V}(X_{lr})],$$

where $X_{lr}$ are the scaled guidance tokens and $M$ is an attention mask enforcing local window and guidance-specific interactions.

---

[1] A VAE downsampling factor of 16 yields $\frac{1024}{16} \times \frac{1024}{16} = 4096$ tokens, similar scaling for other resolutions.

[2] With 16× VAE downsampling, 256×256 windows yield 16×16 tokens, creating attention matrices that efficiently utilize GPU kernels. Smaller window size produces tokens less than the granularity of 128.

We train this framework with the standard flow matching objective, extended to include the low-resolution guidance:

$$L_{\text{FM}}(\theta; C_T, X_{lr}) = \mathbb{E}_{t \sim \mathcal{U}(0,1), X_t \sim p(X|t, C_T, X_{lr})} \left[ \| f_\theta(X_t, t, C_T, X_{lr}) - u_t(X_t; C_T, X_{lr}) \|^2 \right].$$

In our implementation, $X_{lr}$ is generated at $256 \times 256$, while $X$ is trained at $1K \times 1K$. Since the model is merely adapting to a novel attention pattern guided by low-resolution inputs, rather than learning new high-resolution feature, **no native 4K data** are needed. This design yields three key advantages: (1) training is efficient, since only a small set of LoRA parameters is updated; (2) adaptation can be performed entirely on commodity-resolution data; and (3) ultra-high-resolution synthesis is achieved without the need for prohibitively expensive $4K$ training datasets.

**Fused-Kernel Adaptation.** By default, X is reshaped into 1D token sequences by a raster-scan of the 2D latent grid for transformer inputs, which results in a sparse, non-contiguous attention mask that is suboptimal for GPU execution. We address this by re-permuting the tokens in $X$ and $X_{lr}$ in a Hilbert-Curve order as shown in Figure 2 (Figure 15 in the appendix illustrates the effect of this permutation.). This clusters all tokens belonging to a single local window into a dense, contiguous block in memory. We also construct attention mask $M$ for allowed interactions: text attends to itself and $X$ , $X$ attends locally and to both text and $X_{lr}$ regions, $X_{lr}$ attends to text and itself locally. To achieve acceleration, we follow FlashAttention's Dao et al. (2022); Dao (2023) tiling strategy, which allow us to skip computation for entire blocks of the attention matrix masked out. Following Zhang et al. (2025c;e), we adapt the implementation to SageAttention Zhang et al. (2025d;c) yielding additional improvements in speed and VRAM efficiency.

## 4 EXPERIMENTS

### 4.1 IMPLEMENTATION DETAILS

**Experimental Setup.** We adopt FLUX.1-dev Black Forest Labs (2025) as our text-to-image backbone, utilizing the Hugging Face Diffusers library for implementation. Parameter-efficient fine-tuning is conducted using LoRA from the PEFT library, with both rank and alpha equal to 16. LoRA layers are only applied to the Query, Key, and Value projection layers across the DiT blocks. We adopt pyriology scheduler with default learning rate of 1 and weight decay of 0.01, and set bias correction adn safe-guard warmup to true. During inference, we set the NTK factor of RoPE embedding to be 10 and disabled the dynamic shifting of the scheduler to fir the ultra-high resolution generation following Bu et al. (2025); Du et al. (2024b). Our custom attention mechanisms are implemented following the principles of FlashAttention-2 Dao (2023) and SageAttention Zhang et al. (2025c). The model was trained for 20,000 steps on two NVIDIA A6000 Ada GPUs (48GB VRAM each), using a per-GPU batch size of 1 with gradient accumulation over 4 steps. All subsequent experiments were conducted on this same hardware configuration. For the fine-tuning dataset, we generated 10,000 images at a resolution of $1024 \times 1024$ using the base FLUX.1-dev model, prompted by a randomly sampled collection of high-quality text descriptions. The kernel size is designed to be Q-block=128 and K-block=64 (128 for blocksparse Flash Attention) which suits the granularity supported for Zhang et al. (2025c) on NVIDIA A6000 Ada. All experiments were conducted on a cluster of 8 NVIDIA A6000 Ada.

**Metrics and Evaluation Protocol.** To ensure a comprehensive and diverse evaluation, we generated a benchmark suite of 1,000 high-quality prompts across various categories using GPT-4o. We assess performance using a standard set of metrics: CLIP Score Radford et al. (2021) for prompt-image alignment, CLIP-IQA Wang et al. (2023), Fréchet Inception Distance (FID) Heusel et al. (2017), and Inception Score (IS) Salimans et al. (2016) for image quality. The FID score is computed against a reference set of 10,000 real images from the LAION-High-Resolution Schuhmann et al. (2022) dataset. To specifically evaluate the fidelity of local details, we also report patch-based versions of these metrics, $\text{FID}_{\text{patch}}$ and $\text{IS}_{\text{patch}}$, calculated on local image patches. For all comparisons, competing training-free methods were evaluated on the same FLUX.1-dev base model, following their official implementations to ensure fairness.

| Ours | URAE | SANA | HiFlow | I-MAX |
|------|------|------|--------|-------|

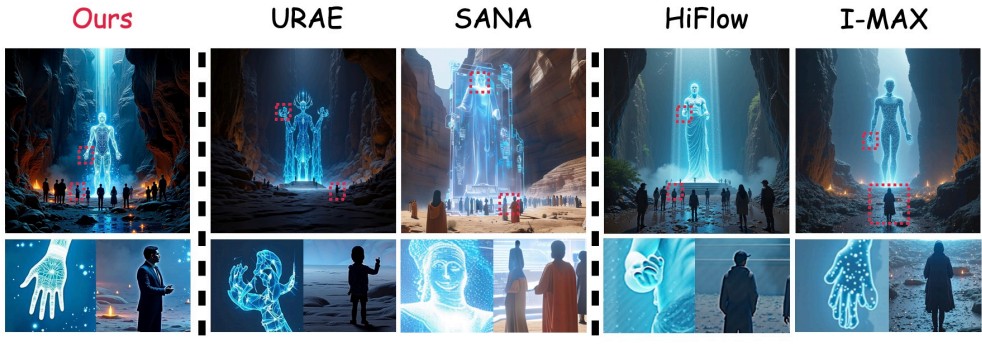

A futuristic AI shrine built in a canyon, worshippers offering digital prayers to holographic deities

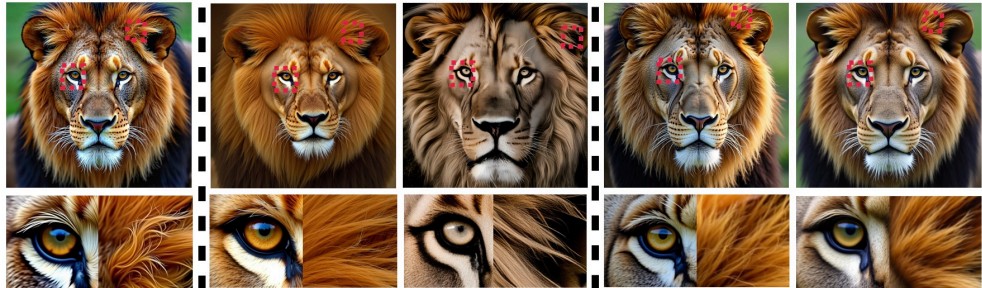

A close-up of a lion's face, majestic mane, intense and regal gaze, king of the savanna, wildlife portraiture

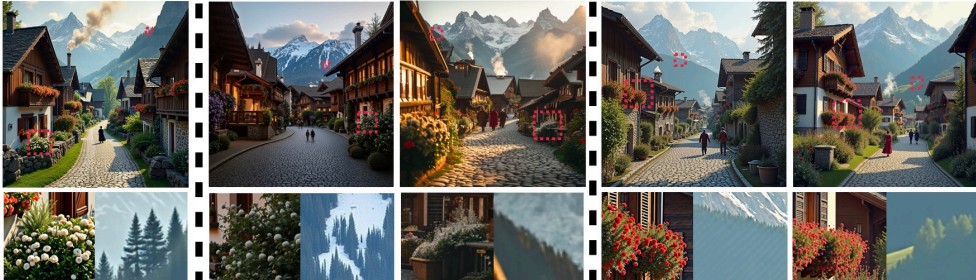

serene mountain village nestled in the Swiss Alps, traditional wooden chalets with flower boxes, cobblestone paths winding between houses, snow-capped peaks in the background, golden hour lighting, smoke rising from chimneys, villagers in traditional clothing walking the streets, cozy warm atmosphere, detailed architecture, rustic charm.

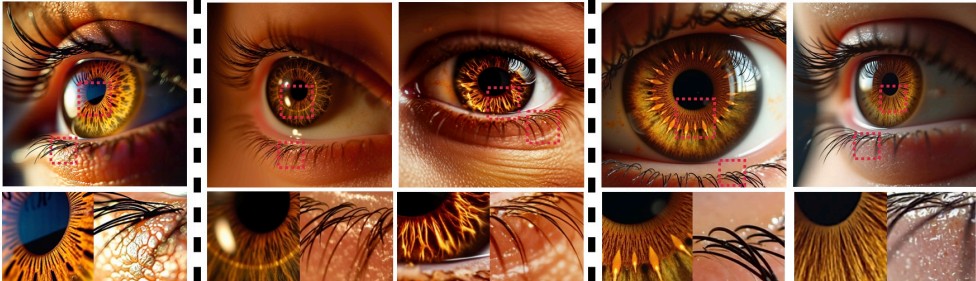

Hyper-realistic macro of amber-brown eye with complex golden iris striations, natural eyelash curl and definition, warm lighting emphasizing copper tones, detailed skin texture around orbital area, ultra sharp focus professional photography

Figure 3: 4K comparison with leading baselines. Zoom in to observe the fine details. **Scale-DiT** produces high-resolution results with superior fidelity. Additional comparisons with other baselines are provided in the Appendix. The middle column shows methods finetuned on 4K data, while the right column shows training-free methods. Revised for better readability.

Table 1: Quantitative comparison at 4K × 4K resolution. The best result is highlighted in bold, while the second-best result is underlined. **Scale-DiT** consistently perform competitive results even compared to training-based methods.

| Method | CLIP-IQA ↑ | FID ↓ | FID$_{patch}$ ↓ | IS ↑ | IS$_{patch}$ ↑ | CLIP Score ↑ |
|---|---|---|---|---|---|---|
| SANA | 0.4457 | 76.31 | 74.27 | 16.68 | **14.02** | 0.3197 |
| Diffusion-4K | 0.3012 | 121.85 | 120.59 | 14.39 | 10.77 | 0.2844 |
| UltraPixel | 0.4421 | 77.42 | 70.94 | 16.98 | 13.26 | **0.3251** |
| URAE | 0.3369 | 67.39 | 62.56 | 17.11 | 12.39 | 0.3204 |
| FLUX+SR (BSRGAN) | 0.3897 | 71.39 | 63.45 | 17.08 | 12.87 | 0.3210 |
| DemoFusion | 0.4392 | 74.89 | 66.37 | 16.23 | 13.02 | 0.3187 |
| DiffuseHigh | 0.4221 | 81.54 | 73.35 | 16.15 | 13.08 | 0.3175 |
| HiDiffusion | 0.4021 | 172.46 | 217.49 | 9.43 | 8.12 | 0.3129 |
| FreCas | 0.2704 | 213.44 | 322.08 | 7.98 | 6.84 | 0.2805 |
| I-MAX | 0.4381 | 70.33 | 65.67 | 16.50 | 12.69 | 0.3211 |
| HiFlow | 0.4407 | 69.18 | 63.72 | 17.13 | 13.43 | 0.3113 |
| **Scale-DiT (Ours)** | **0.4505** | **67.03** | **61.78** | **17.21** | 13.31 | 0.3231 |

Table 2: Performance of our method from 1K to 4K on the GenEval benchmark. Results show that output quality is maintained across resolutions despite training on 1K data, demonstrating scalability and resolution-agnostic design.

| Model | Overall | 1 Obj | 2 Obj | Counting | Colors | Position | Attr. Binding |
|---|---|---|---|---|---|---|---|
| FLUX.1-Dev | 0.66 | 0.98 | 0.81 | 0.74 | 0.79 | 0.22 | 0.45 |
| Ours(2K) | 0.67 | 0.99 | 0.83 | 0.74 | 0.81 | 0.22 | 0.45 |
| Ours(3K) | 0.67 | 0.98 | 0.82 | 0.73 | 0.81 | 0.23 | 0.46 |
| Ours(4K) | 0.67 | 0.98 | 0.83 | 0.75 | 0.79 | 0.24 | 0.45 |

## 4.2 COMPARISON WITH STATE-OF-THE-ART METHODS

We compare our results with methods that can also generate 4K resolution images. These include data-intensive, training-based methods: SANA Xie et al. (2024), Diffusion-4K Zhang et al. (2025b), FLUX+URAE Yu et al. (2025), and UltraPixel Ren et al. (2024); a super-resolution baseline: FLUX+BSRGAN Zhang et al. (2021); and several training-free approaches: DemoFusion Du et al. (2024a), DiffuseHigh Kim et al. (2025), I-MAX Du et al. (2024b), and HiFlow Bu et al. (2025).

**Qualitative Comparison.** Figure 1 presents qualitative results of **Scale-DiT** at $4K \times 4K$, $2K \times 3K$, and $2K \times 2K$, demonstrating superior text-to-image generation with high-fidelity details and coherent global structure. A more detailed comparison at 4K is shown in Figure 3, where we include a representative subset of leading baselines for clarity; comprehensive comparisons against the nine methods, as well as additional 2K and 4K results, are provided in the Appendix. At 4K resolution, our method renders anatomically correct hands with exceptional detail as shown in the first row, while SANA produces a blurry image, HiFlow introduces diagonal artifacts, and I-MAX fails to generate the correct number of fingers. In the second and third row, **Scale-DiT** consistently preserves textural clarity, such as fur and tree structures, surpassing competitors while maintaining global coherence. The fourth row highlights highly detailed eye generation at 4K, where our method clearly captures iris structure, eyelashes, and skin texture with remarkable realism. These examples collectively demonstrate **Scale-DiT**'s ability to synthesize images that combine fine-grained local details with coherent large-scale composition, validating its performance across diverse high-resolution scenarios.

**Quantitative Comparison.** Table 1 presents a quantitative comparison of our method against the nine state-of-the-art methods at 4K resolution (please refer to the Appendix for 2K comparisons). The results show that our method achieves highly competitive performance across all metrics, even outperforming training-based methods that rely on extensive 4K-resolution training data. Table 2 demonstrates consistent performance on the GenEval Ghosh et al. (2023) benchmark, confirming that our method's output quality is resolution-agnostic.

**Efficiency and Scalability Analysis.** The efficiency gains of our approach are summarized in Table 3 and Figure 4. Focusing on methods built upon the FLUX.1-dev backbone for a fair comparison, our method achieves a $1.5\times$ to $2.7\times$ inference speedup over its counterparts (with all $> 2.4\times$ speedup per denoising iteration). In terms of memory usage, our method is highly efficient: excluding the base model's footprint ( 33 GB), it requires only 2.8 GB of additional VRAM, whereas

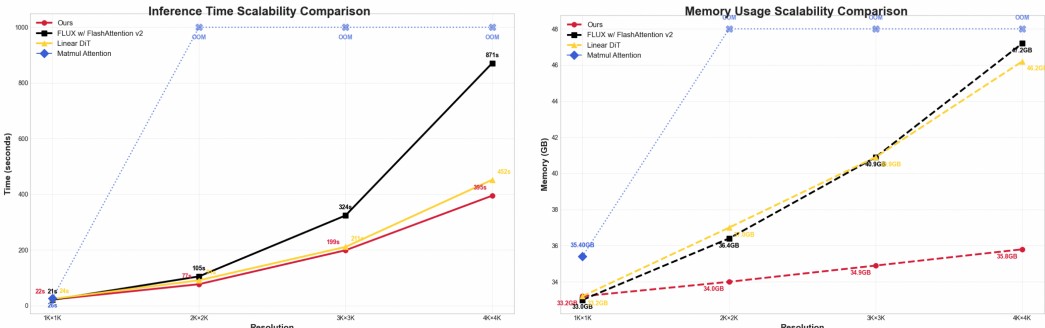

Figure 4: Memory and speed statistics when scaling from 1K to 4K comparing with native FLUX.1 Dev with FlashAttention-2 (black line), Linear Attention used in SANA (yellow line), and naive matmul attention (blue line). Our method's (in red) advantage in efficiency is more and more significant as resolution increases comparing to the others. OOM stands for out of memory. (Please check Figure 13 in the appendix for clearer zoomed-in figures.)

a naive implementation with FlashAttention consumes 14.1 GB, making our approach a lot more memory-efficient, which enables inferencing on smaller consumer-level GPUs, (i.e. 40 GB A100), and more GPU friendly finetuning. Figure 4 further illustrates scalability. As the target resolution increases from 1K to 4K, the gap in both latency and memory usage between our method and the dense attention baseline using FlashAttention2 Dao (2023) widens significantly.

Compared with Linear Attention, our approach remains faster and more memory-efficient. Linear attention, while computationally faster than FlashAttention, requires scaling the hidden dimension with sequence length to maintain performance Han et al. (2024); Meng et al. (2025); Zhang et al.. In contrast, our method uses local window attention, which does not need to scale with resolution, enabling consistent memory and latency efficiency as resolution increases. Since linear attention requires the hidden dimension to scale with resolution, previous works would need to fine-tune on high-resolution data for each target resolution, whereas our framework achieves ultra-high-resolution synthesis using only 1K-resolution training data without any performance drop. Collectively, these results validate the scalability, efficiency, and robustness of our proposed framework.

Table 3: Comparison of memory (GB) and latency (sec; time to generate one 4K latents (65536 tokens)) across FLUX.1-dev-based 4K generation methods. Our method achieves $2\times$ speedup and significant memory reduction (given that the base model itself is around 33 GB in size).

| Method | Ours | FLUX | HiFLOW | URAE | I-MAX |
|---|---|---|---|---|---|
| Memory(GB) | 35.8 | 47.2 | 43.4 | 41.3 | 43.9 |
| Latency(sec) | 395 | 871 | 634 | 823 | 1091 |
| Seconds per iteration(sec/iter) | 13.25 | 31.11 | 31.59 | 29.40 | 35.63 |

## 5 ABLATION STUDY

To validate our core design choices, we conduct a series of ablation studies analyzing the impact of window size, guidance resolution ratio, attention patterns, and token permutation strategies.

Table 4: Ablation experiments investigating the effects of window size and the high-to-low resolution ratio demonstrated a negligible impact on model performance. Therefore, to prioritize computational efficiency, the configuration utilizing the smallest window size and the largest ratio was chosen for all subsequent experiments.

| Method | ws256 $\rho4$ | ws256 $\rho2$ | ws512 $\rho4$ | ws1024 $\rho4$ |
|---|---|---|---|---|
| FiD $\downarrow$ | 67.23 | 67.19 | 66.31 | 67.83 |
| FiD$_{patch}$ $\downarrow$ | 61.46 | 62.28 | 63.48 | 62.61 |

**Window size and resolution ratio.** We conduct ablation study on window size (ws) and ratio between high-res and low-res image ($\rho$). We experiment window sizes from 256, 512, and 1024, and $\rho = 2, 4$. As shown in Table 4, the variations in FiD scores across these configurations were not statistically significant. To maximize computational efficiency without compromising performance,

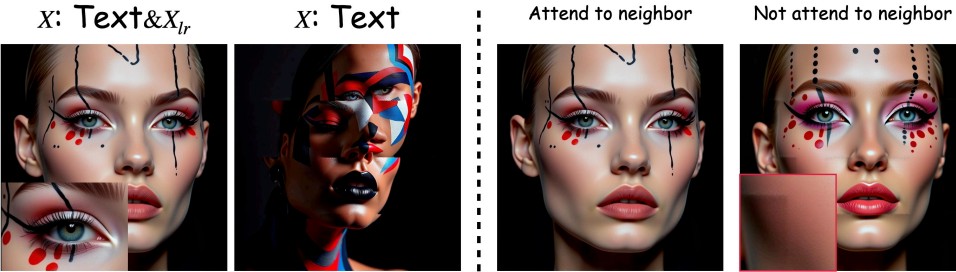

Figure 5: Ablation Study on attention scale. Images from left to right corresponds to cases that 1) High-res $X$ attends to both text and $X_{lr}$. 2) $X$ attends to only Text. 3) Each window of $X$ attends to its neighbors. 4) $X$ only attends to itself. 1) and 2) demonstrates the effectiveness of our low-res guidance, and 3) and 4) illustrates that with only allowing each window to attend to part of the neighbors will solve the boundary issues.

we selected the smallest window size (ws=256) and the largest ratio ($\rho$=4). It is worth noting that smaller window sizes or larger ratios are constrained by the fixed granularity of the kernel which depends on the hardware design of the GPU.

**Attention Pattern** We conducted an ablation study to evaluate the impact of different text-to-image attention mechanisms. Case 1 examines the effect of attending to low-resolution tokens $X_{\mathrm{lr}}$, comparing (a) high-resolution tokens $X$ attending to both textual embeddings and $X_{\mathrm{lr}}$ versus (b) attending only to textual embeddings. Case 2 investigates the effect of attending to neighboring windows, comparing (a) each window of $X$ attending to its neighbors versus (b) attending only to itself.

As shown in Figure 5, including low-resolution tokens in Case 1 produces consistently stable results, whereas excluding them causes discontinuities and multi-face artifacts, highlighting the importance of low-resolution guidance. In Case 2, attending to neighboring windows effectively eliminates boundary artifacts, while incurring only a minimal additional computational cost of less than 2%.

**Discussion on whether we need 4K native data.** Our method calculates attention within local windows while maintaining global information through low-resolution representations, allowing efficient and high-quality 4K generation without requiring 4K training data. We do not deny that real 4K data can provide richer fine-scale details absent from 1K pretraining. We believe that the global and semantic structures of generated images are captured in the latent space, while high-resolution details are stored in the VAE encoder and decoder. To verify this hypothesis, we replace the Flux VAE with the UltraFlux VAE Ye et al. (2025), which post-trains the Flux VAE using 4K data. This replacement yields clear enhancements in high-resolution details (see Figure 14 in the appendix), confirming that our approach effectively separates global structure modeling from fine-detail generation. Overall, our framework demonstrates that 1K training data are sufficient for latent-space 4K generation, while real 4K data serve as a complementary boost for detail refinement.

## 6 CONCLUSION

We present **Scale-DiT**, a novel framework enabling pre-trained diffusion transformers to generate ultra-high-resolution images *without requiring additional high-resolution training data*. **Scale-DiT** introduces a hierarchical attention mechanism that partitions high-resolution latents into fixed-size local windows while maintaining global coherence through low-resolution guidance with scaled positional anchors. The framework comprises three key components: (i) efficient local window attention that reduces quadratic complexity to near-linear scaling; (ii) global semantic preservation via low-resolution guidance latents with position scaling; and (iii) parameter-efficient joint denoising through LoRA-based adaptations trained solely on commodity resolutions. Extensive experiments demonstrate that **Scale-DiT** achieves superior visual quality at 4K resolution while delivering over $2\times$ *speedup* and *less memory usage* compared to dense attention baselines, establishing a practical paradigm for scalable ultra-high-resolution text-to-image generation.

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

# APPENDIX

## A    4K GALLERY

In Figure 6, 7, 8 we present more 4K resolution results. More results are in the supplementary materials.

## B    2K GALLERY

In Figure 9, we present more 2K resolution results. More results are in the supplementary materials.

## C    LLM USAGE DECLARATIONS

We declare that Large Language Models (LLMs) were used in limited capacity during the preparation of this manuscript. Specifically, LLMs were employed for: (i) generating diverse text prompts for model evaluation and dataset creation, (ii) grammar checking and language refinement of the manuscript, and (iii) assisting in data collection procedures for experimental validation. All core technical contributions, experimental design, analysis, and conclusions presented in this work are entirely ours own. The use of LLMs did not influence the scientific methodology, results interpretation, or theoretical contributions of this research.

## D    REPRODUCIBILITY STATEMENT

We will release our source code upon acceptance.

## E    COMPLETE QUALITATIVE COMPARISON

In figure 10 and figure 11, we present the complete qualitative comparison of 4K resolution images with the quantitatively compared baselines in Table 1. 2K resolution comparison are demonstrated in Figure 12. We revised the figure for better readability.

## F    QUANTITATIVE COMPARISON ON 2K RESOLUTION

Table 5 demonstrates the quantitative results for $2K \times 2K$ resolution image generation. Our method is in leading position across all metrics.

Table 5: Quantitative comparison at 2K × 2K resolution. The best result is highlighted in bold, while the second-best result is underlined.

| Method | FID ↓ | FID$_{patch}$ ↓ | IS ↑ | IS$_{patch}$ ↑ | CLIP Score ↑ |
|--------|-------|-----------------|------|----------------|--------------|
| SANA | 75.79 | 74.26 | 20.15 | 15.21 | 0.3188 |
| Diffusion-4K | 103.22 | 97.10 | 19.88 | 13.79 | 0.3159 |
| UltraPixel | 75.51 | 72.97 | 21.31 | 16.12 | 0.3148 |
| URAE | **66.24** | 62.07 | 21.53 | **17.32** | 0.3214 |
| FLUX+SR (BSRGAN) | 74.38 | 72.84 | 21.07 | 16.89 | 0.3201 |
| DemoFusion | 78.69 | 72.45 | 20.93 | 15.14 | 0.3197 |
| DiffuseHigh | 71.69 | 65.58 | 20.57 | 15.32 | 0.3132 |
| I-MAX | 67.39 | 62.78 | 21.43 | 17.09 | 0.3204 |
| HiFlow | 67.84 | 62.06 | 21.68 | 16.94 | **0.3235** |
| **Scale-DiT (Ours)** | 67.23 | **61.46** | **21.72** | 17.26 | 0.3216 |

## G    SUPPLEMEMTARY FIGURES FOR THE MAIN PAPER.

Figure 15 and Figure 14 demonstrate the figures for the sections 5 and 3.

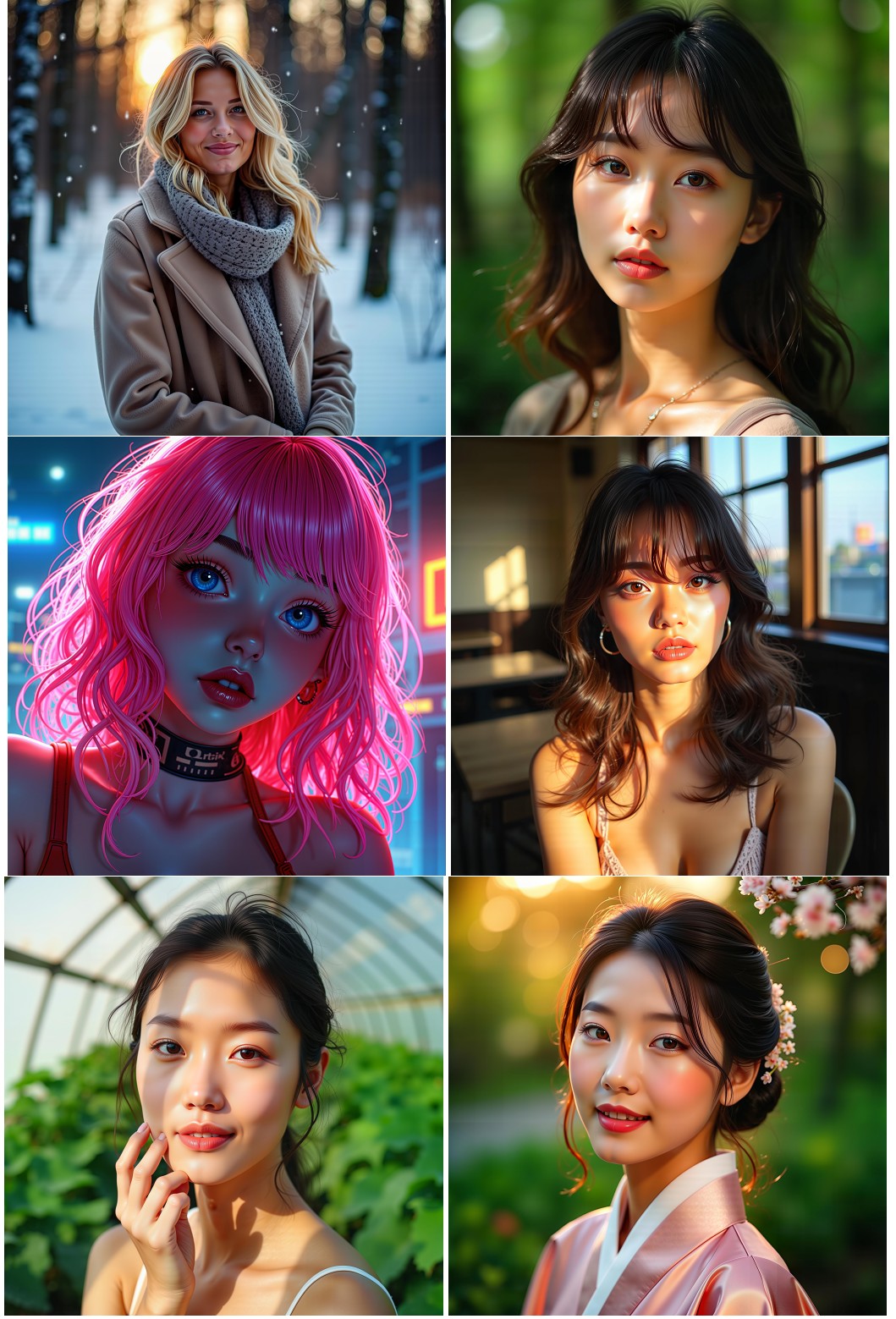

Figure 6: More 4K results

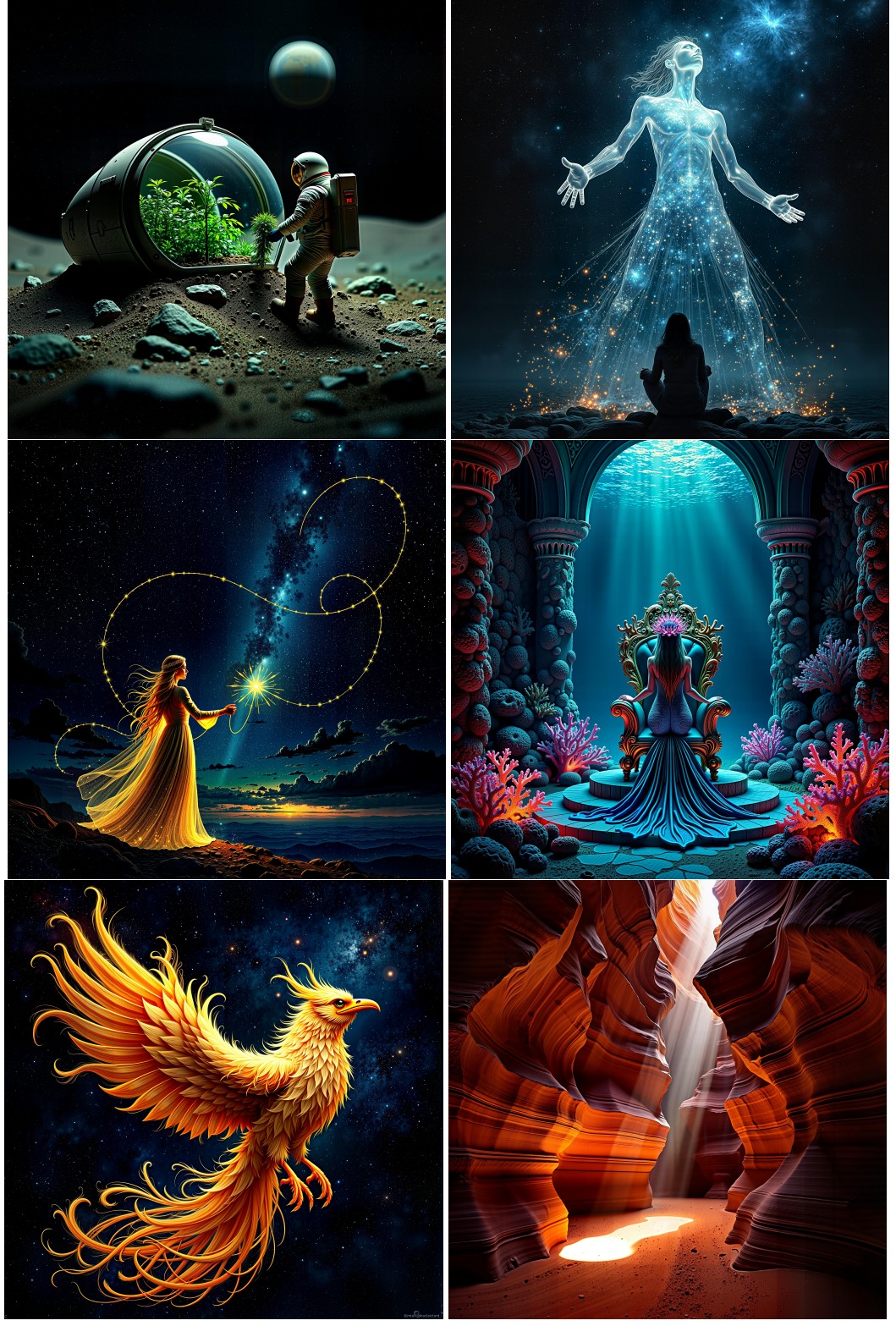

Figure 7: More 4K results

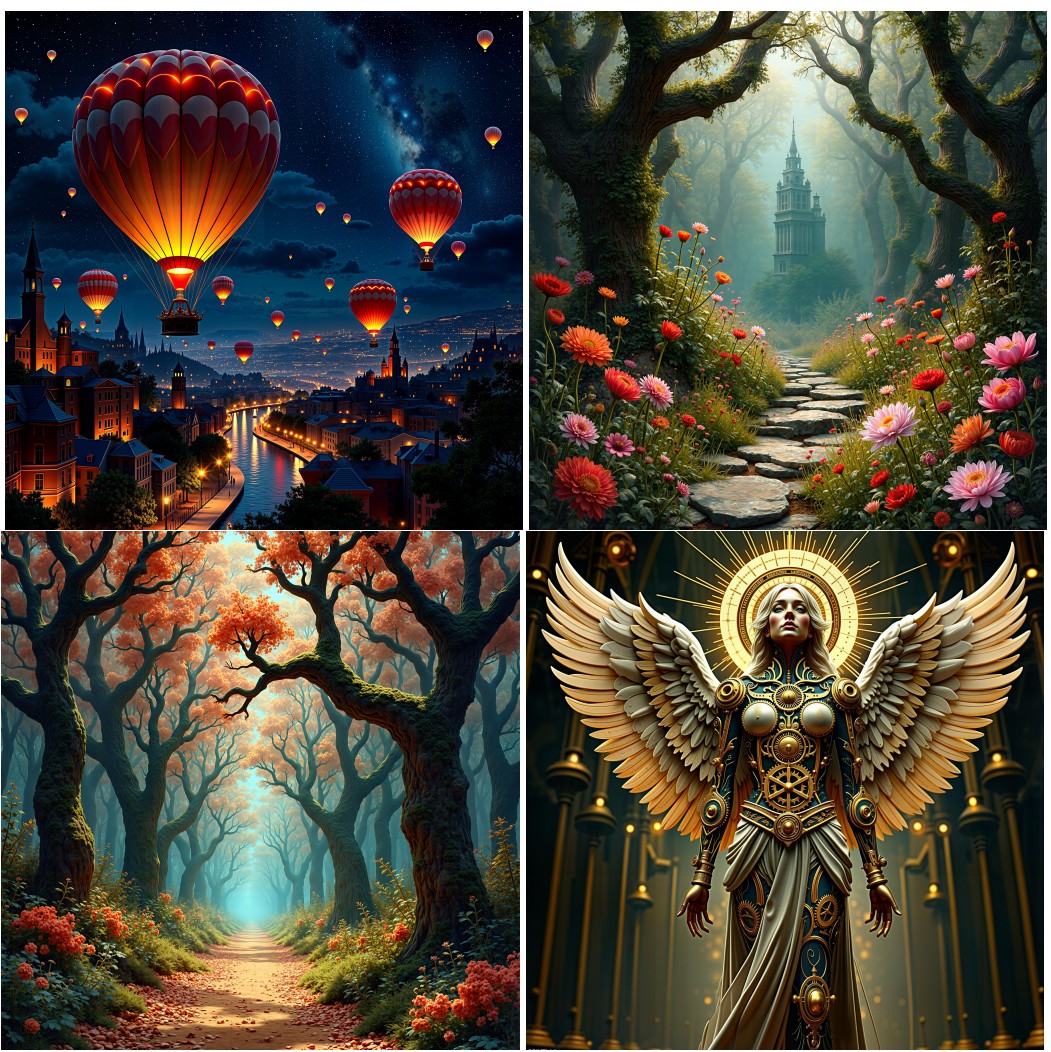

Figure 8: More 4K results

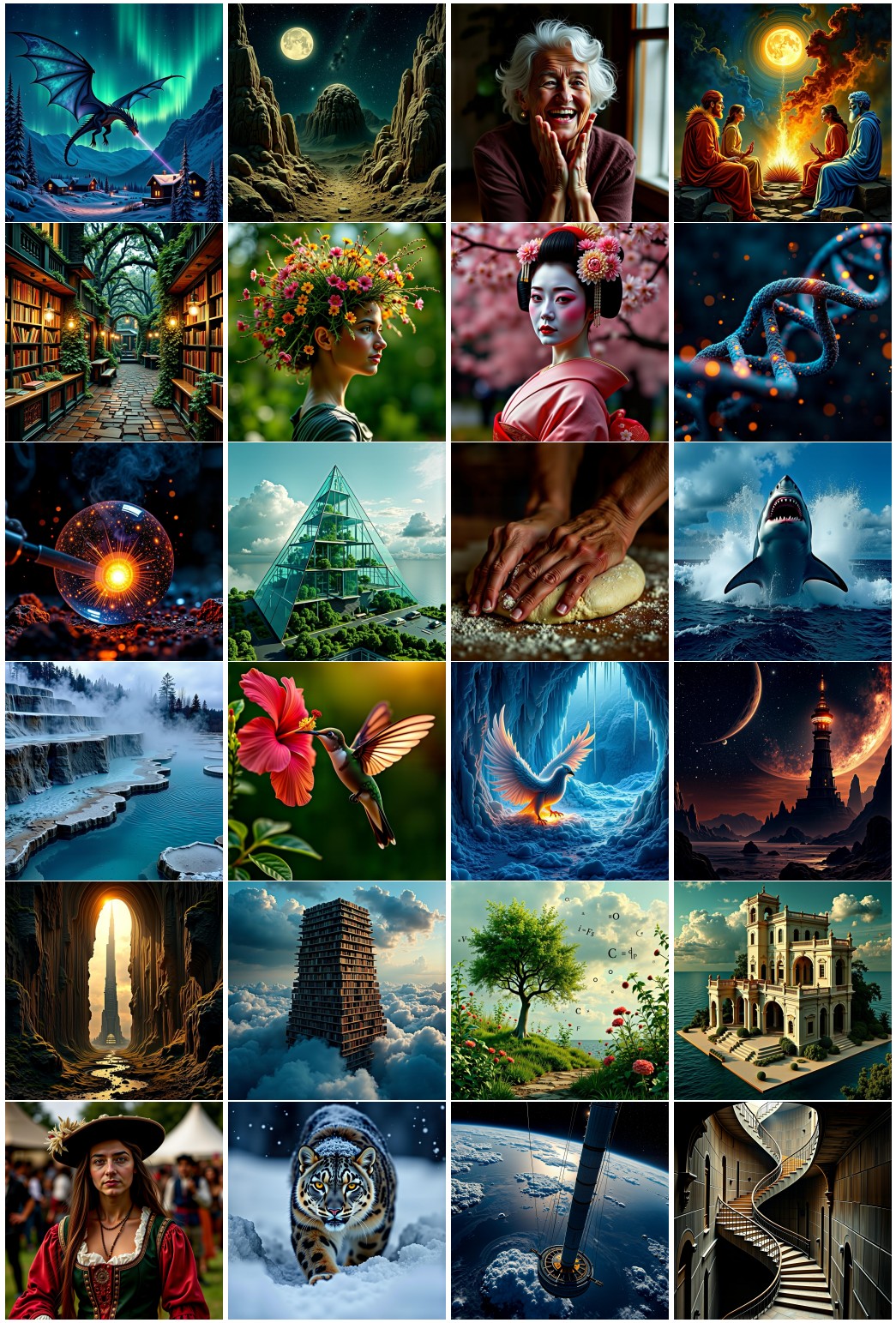

Figure 9: More 2K results.

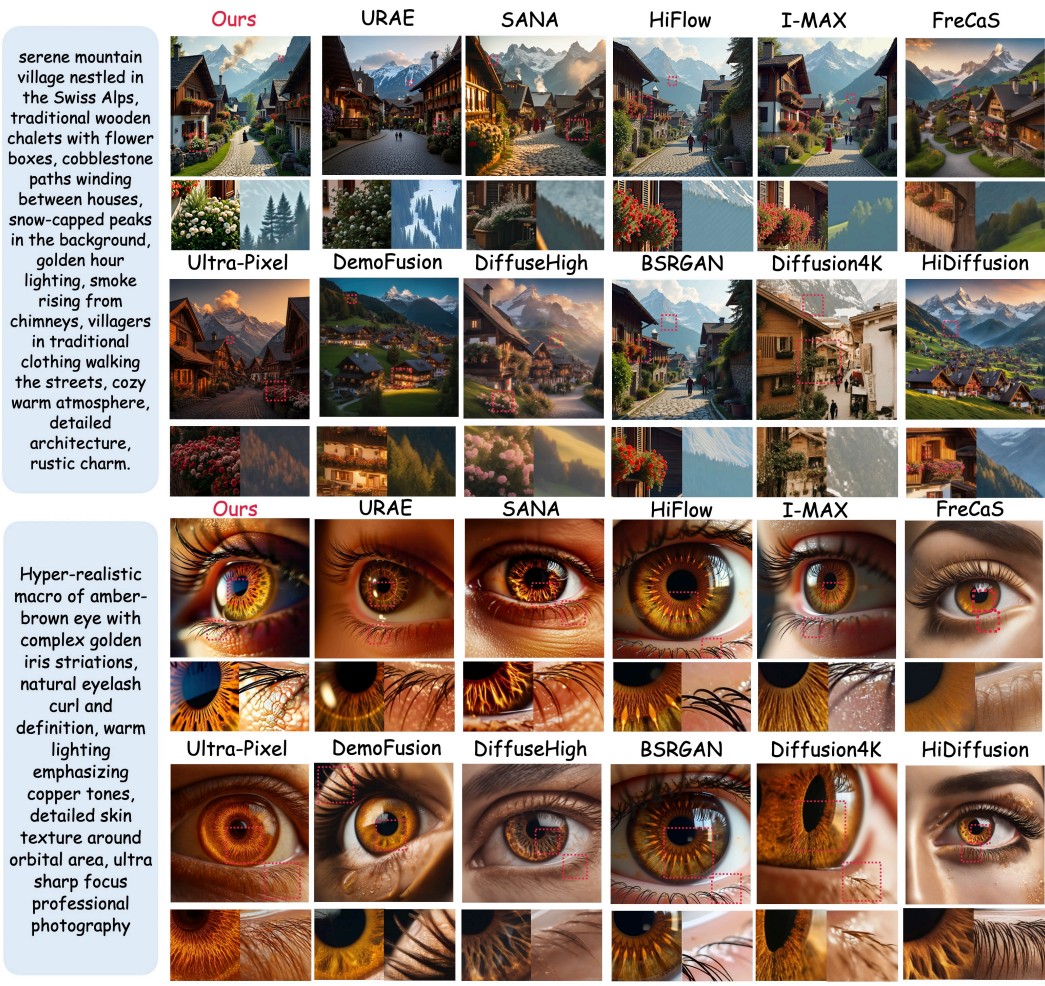

Figure 10: Complete comparison against the 9 baselines on 4K resolution. Zoom in to view the details, our method produces the best results. Revised for better readability.

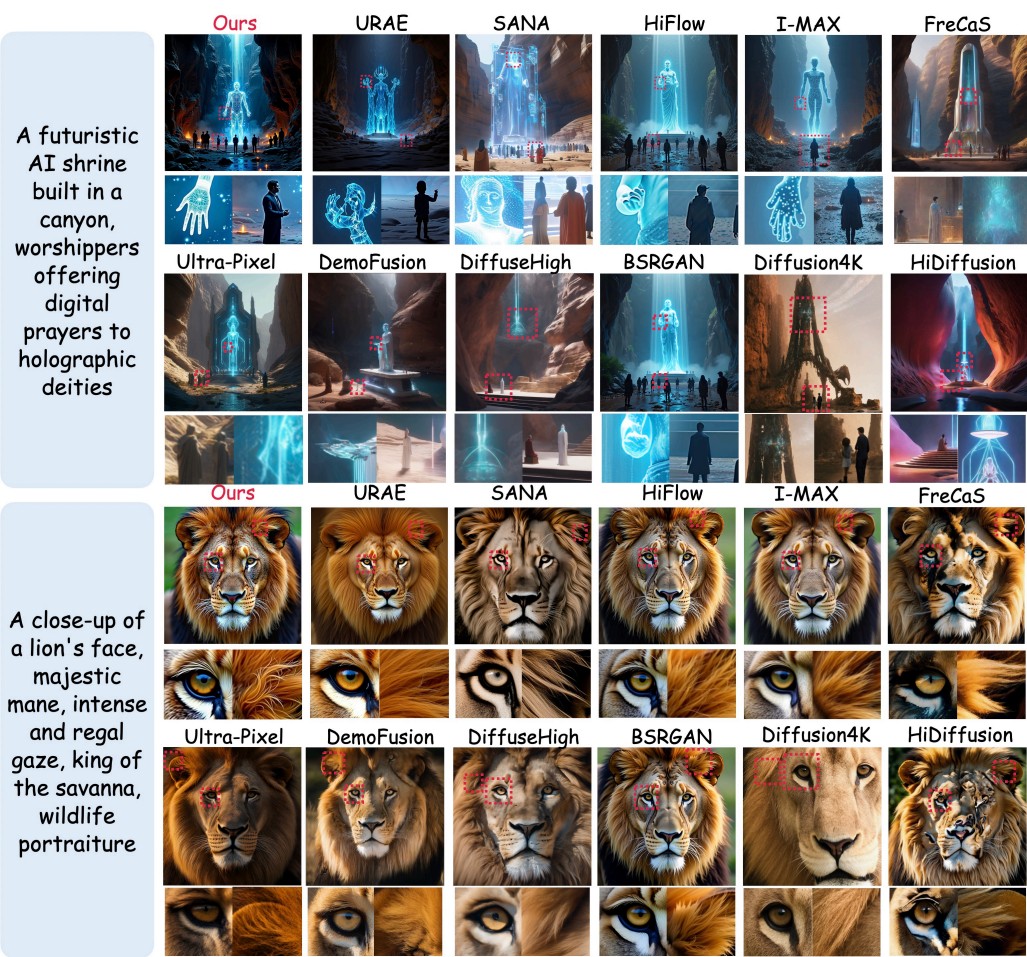

Figure 11: Complete comparison against the 9 baselines on 4K resolution. Zoom in to view the details, our method produces the best results. Revised for better readability.

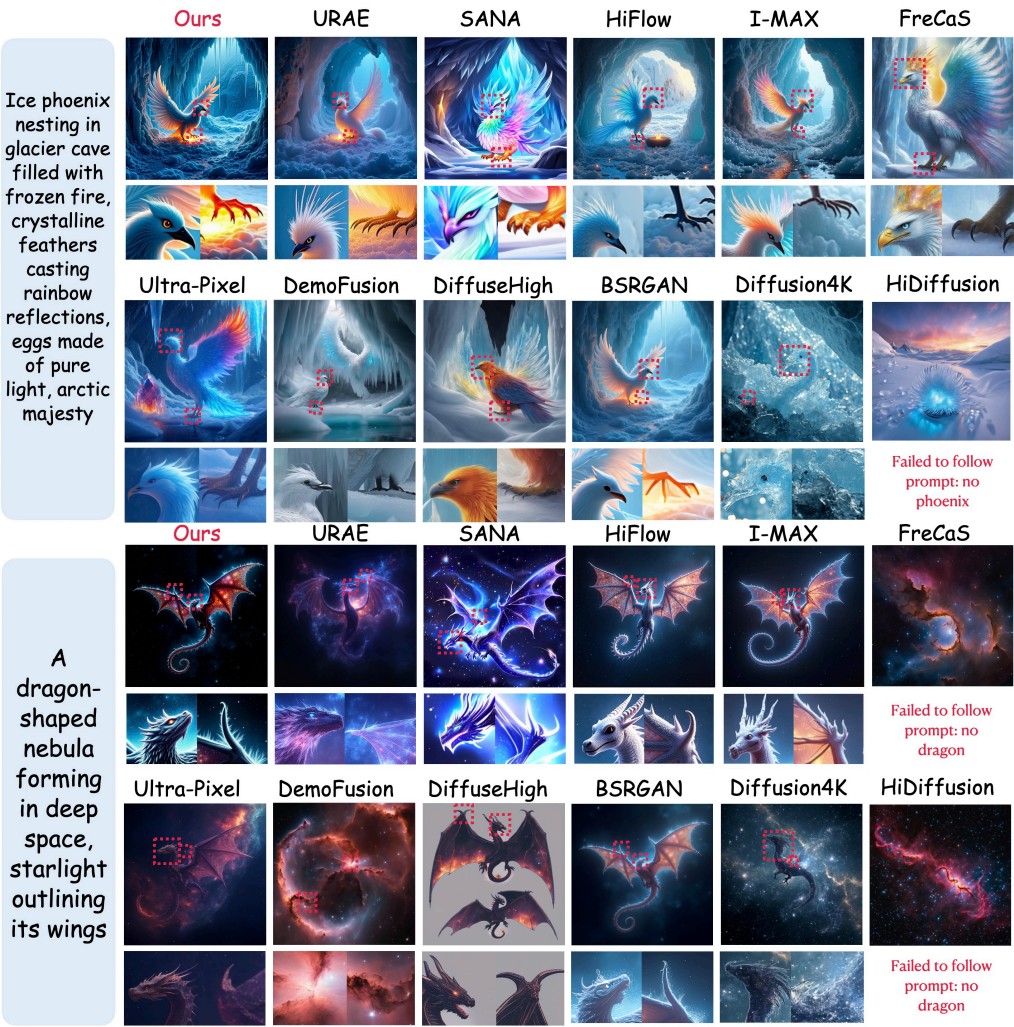

Figure 12: Complete comparison against the 9 baselines on 2K resolution. Zoom in to view the details, our method produces the best results. Revised for better readability.

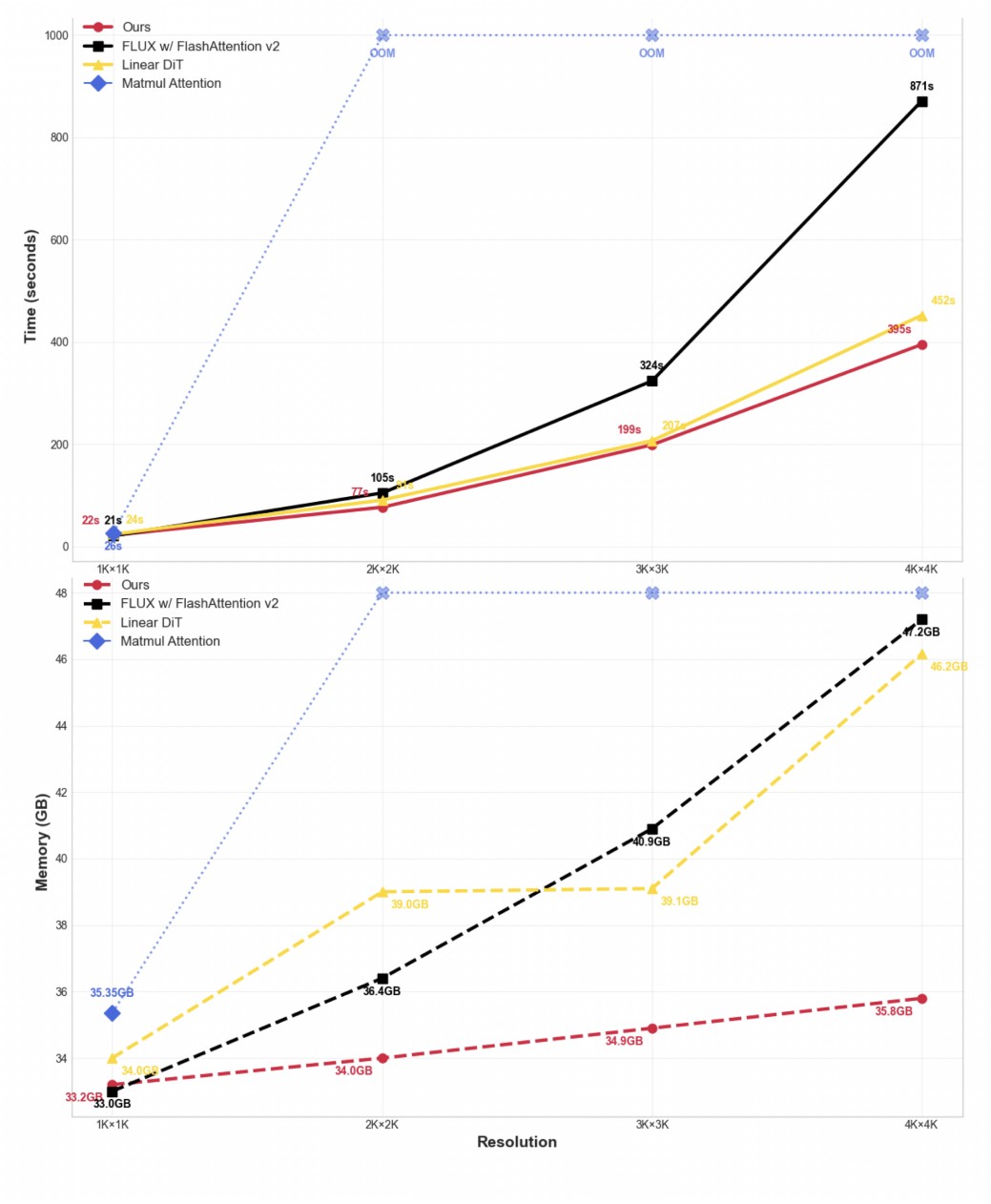

Figure 13: Zoomed in figure of scaling performance. We resolution scales up our model's advantage in speed and memory usage is higher and higher.

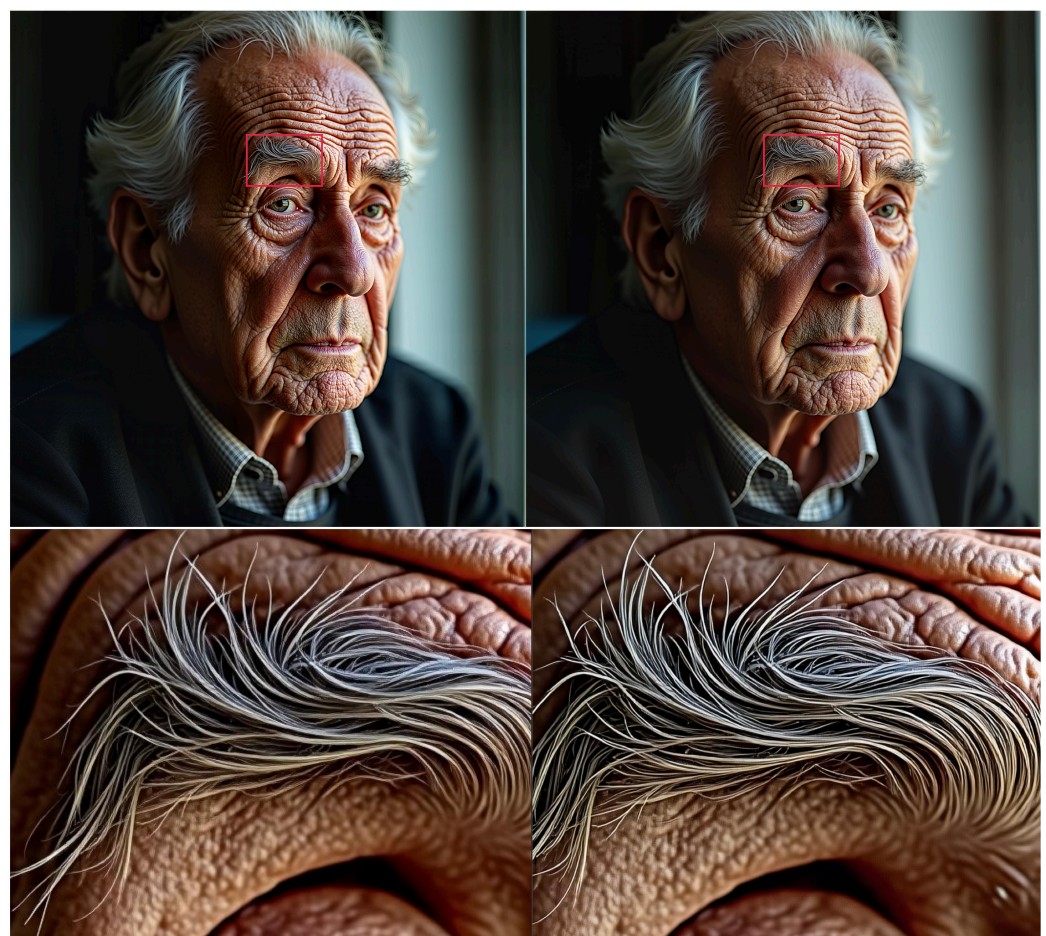

Figure 14: Figure demonstrate the quality improvement after using 4K finetuned VAE. We can see clearer details in the eyebrow. The left image is the one without 4K finetuning, the right image is with 4K finetuning. Please refer to section 5 for more descriptions.

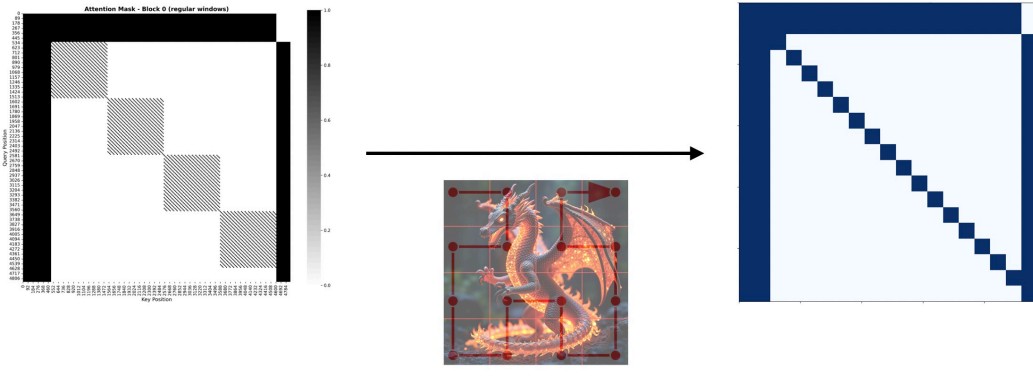

Figure 15: Figure demonstrates the advantage of the usage of Hilbert order permutation, we have a dense contiguous layout for acceleration. Please refer to "Fused-kernel Adaptation" subsection 3 in the methodology section

# H  DISCUSSION ON ADVANTAGES OF LOCAL ATTENTION

## H.1  LINEAR COMPUTATIONAL SCALABILITY

Our local window attention achieves **linear scaling** with respect to image area, fundamentally different from the quadratic complexity of full attention. For an image of size $H \times W$ with local windows of size $w \times w$:

$$\text{Computational Complexity} = \frac{H \times W}{s^2 \times w^2} \times w^4 = O(HW) \tag{1}$$

where $s$ is the VAE downsampling factor. This linear relationship means that doubling the image resolution doubles the computational cost, rather than quadrupling it as in full attention systems. The computational density (operations per pixel) remains constant:

$$\rho = \frac{\text{Total Operations}}{H \times W} = \frac{w^2}{s^2} = \text{constant} \tag{2}$$

This property enables practical scaling to arbitrarily large resolutions with predictable resource requirements.

## H.2  QUALITY PRESERVATION ACROSS RESOLUTIONS

A critical advantage of our approach is that **representational quality remains constant** regardless of output resolution. Since every local window operates on the same relative position range $\mathcal{R}_{\text{local}} = \{(\Delta i, \Delta j) : |\Delta i|, |\Delta j| \leq w - 1\}$, and this range falls entirely within the pretrained distribution $\mathcal{D}_{\text{train}}$, each window achieves identical quality.

The per-unit quality across the entire image is:

$$\text{Quality per unit area} = \frac{Q(\mathcal{R}_{\text{local}})}{w^2 \times s^2} = \text{constant} \tag{3}$$

This theoretical guarantee means that a 4K image generated by our method has the same quality as a 1K image, unlike approaches that suffer quality degradation when extrapolating beyond training distributions. The quality is **resolution-invariant** because the fundamental building blocks (local spatial relationships) remain within the learned parameter space.

## H.3  NUMERICAL STABILITY THROUGH BOUNDED ATTENTION

Local window attention maintains **numerical stability** by operating within bounded attention dimensions. While full attention at high resolutions requires increasingly high precision, our approach maintains constant precision requirements.

The information content per attention weight scales as:

- **Full attention**: $-\log\left(\frac{s^2}{HW}\right) = \log(HW) - 2\log(s) \to \infty$ as resolution increases
- **Local attention**: $-\log\left(\frac{1}{w^2}\right) = 2\log(w) = \text{constant regardless of resolution}$

This fundamental difference means that local attention avoids the **attention dilution problem** where each query token must distribute probability mass across tens of thousands of key tokens, leading to numerical precision issues and gradient vanishing. Instead, each query attends to only $w^2$ tokens, maintaining stable softmax distributions and reliable gradient flow.

The bounded nature of local attention ensures that the method remains **numerically robust** at any resolution, while full attention systems become increasingly unstable as they approach hardware precision limits.

