# OpenReview forum: "Scale-DiT: Ultra-High-Resolution Image Generation with Hierarchical Local Attention"
_ICLR.cc/2026/Conference — Submitted to ICLR 2026_

### Official Review · Reviewer_nPgv · 2025-10-29

**Soundness:** 3
**Presentation:** 2
**Contribution:** 3
**Rating:** 6
**Confidence:** 5

**Summary:**

This paper presents an efficient framework for ultra-high-resolution image generation. Specifically, it uses a standard-resolution image as structure guidance and trains LoRA-based attention layers to attend high-resolution tokens to their counterparts in the low-resolution image. To accelerate practical inference, the paper masks most irrelevant tokens and permutes the token order using Hilbert scanning. Experiments demonstrate the effectiveness and efficiency of the proposed method.

**Strengths:**

1. The proposed method is straightforward and somewhat novel.
2. The experiments demonstrate the effectiveness of the proposed method

**Weaknesses:**

1. The figures are not well organized, and the text in nearly all of them is too small to read clearly. This significantly hinders the readability and overall presentation quality.

2. Some related works are missing. For example, HiDiffusion [1] and FreCaS [2] are two recent methods specifically designed for efficient ultra-high-resolution image generation, and a comparison or discussion with these approaches would strengthen the paper’s positioning and context.

3. The training settings for LoRA are not provided. It would be beneficial to report key details such as the rank, layers involved, and training schedule, to ensure reproducibility and clarity.

[1] HiDiffusion: Unlocking High-Resolution Creativity and Efficiency in Low-Resolution Trained Diffusion Models, ECCV 2024.
[2] FreCaS: Efficient Higher-Resolution Image Generation via Frequency-aware Cascaded Sampling, ICLR 2024.

**Questions:**

see weakness

---

> ### Author Response · Authors · 2025-11-28
>
> ### W1:
> *“The figures are not well organized, and the text in nearly all of them is too small to read clearly. This significantly
> hinders the readability and overall presentation quality.”*
>
> **Response**: Thanks. We will update them in the revised version.
>
> ---
>
> ### W2:
> *“Some related works are missing. For example, HiDiffusion and FreCaS are two recent methods specifically designed
> for efficient ultra-high-resolution image generation, and a comparison or discussion with these approaches would
> strengthen the paper’s positioning and context.”*
>
> **Response**:
> Both papers are now included in our references, and we added a short discussion in the revised related work
> section. HiDiffusion and FreCaS target UNet-based architectures: HiDiffusion is training-free and regularizes the
> denoising trajectory but still struggles with long-range dependencies, whereas our method explicitly decouples fine-
> detail generation from global semantic reasoning using local-window attention and low-resolution guidance. FreCaS
> uses a coarse-to-fine strategy to synthesize only high-frequency content, which reduces the long-range dependency
> burden but requires careful stage alignment and CFG tuning. In contrast, our approach maintains global consistency
> without multi-stage repair or delicate parameter balancing. To quanitatively compare the performance, we included
> their results into the Table 1 of the main paper (also Table 2 below here), and our results are still better.
>
> #### Table 2: Completed quantitative comparison at 4K × 4K resolution. The best result is highlighted in bold, while the second-best result is underlined. Scale-DiT consistently performs competitive results even compared to training-based
> methods.
>
> | Method | FID ↓ | FIDpatch ↓ | IS ↑ | ISpatch ↑ | CLIP Score ↑ |
> |---|---|---|---|---|---|
> | SANA | 76.31 | 74.27 | 16.68 | 14.02 | 0.3197 |
> | Diffusion-4K | 121.85 | 120.59 | 14.39 | 10.77 | 0.2844 |
> | UltraPixel | 77.42 | 70.94 | 16.98 | 13.26 | 0.3251 |
> | URAE | 67.39 | 62.56 | 17.11 | 12.39 | 0.3204 |
> | FLUX+SR (BSRGAN) | 71.39 | 63.45 | 17.08 | 12.87 | 0.3210 |
> | HiDiffusion | 172.46 | 217.49 | 9.43 | 8.12 | 0.3129 |
> | FreCas | 213.44 | 322.08 | 7.98 | 6.84 | 0.2805 |
> | DemoFusion | 74.89 | 66.37 | 16.23 | 13.02 | 0.3187 |
> | DiffuseHigh | 81.54 | 73.35 | 16.15 | 13.08 | 0.3175 |
> | I-MAX | 70.33 | 65.67 | 16.50 | 12.69 | 0.3211 |
> | HiFlow | 69.18 | 63.72 | 17.13 | 13.43 | 0.3113 |
> | Scale-DiT (Ours) | 67.03 | 61.78 | 17.21 | 13.31 | 0.3231 |
> ---
>
> ### W3:
> *“The training settings for LoRA are not provided. It would be beneficial to report key details such as the rank, layers
> involved, and training schedule, to ensure reproducibility and clarity.”*
>
> **Response**: We thank the reviewer for the reminder and have included this in the revised version. Specifically, we
> set LoRA rank and alpha to 16, applying LoRA only to the QKV projection layers across all blocks. We use the
> Pyriology scheduler with a default learning rate of 1 and weight decay of 0.01, with bias correction and safe-guard
> warmup enabled. During inference, we set the NTK factor of ROPE to 10 and disable dynamic scheduler shifting
> to accommodate ultra-high-resolution generation.

---

### Official Review · Reviewer_SUzV · 2025-11-01

**Soundness:** 3
**Presentation:** 2
**Contribution:** 3
**Rating:** 6
**Confidence:** 4

**Summary:**

This paper proposes Scale-DiT, a new diffusion framework designed for efficient and effective generation of ultra-high-resolution images. The core ideas are hierarchical local attention and low-resolution global guidance to generate images without needing native high-resolution training data. The approach involves fine-tuning only the multi-modal attention layers using LoRA, significantly reducing training costs. Experiments confirm that this approach achieves notable efficiency, including reduced memory usage and faster inference speeds compared to baselines.

**Strengths:**

- The method effectively generates ultra-high-resolution images, which is impressive.
- Mixing low-res and high-res data for training is an interesting idea and appears to be very effective.
- The training cost is very low due to the efficient LoRA fine-tuning, requiring tuning only on the mm-attention modules.
- Experimental results clearly demonstrate the method's efficiency advantages, showing significantly lower memory usage and higher generation speeds compared to baseline methods.

**Weaknesses:**

- The quantitative evaluation could be more comprehensive. Currently, the paper only includes FID and IS metrics, which alone aren't sufficient to fully capture the image quality. Including additional Image Quality Assessment (IQA) metrics such as PSNR or CLIP-IQA would provide a more complete evaluation.
- In terms of visual comparisons (e.g., Fig. 3 on page 6), Scale-DiT does not show clear visual superiority, especially when compared against the baseline HiFlow. The generated images from Scale-DiT are good, but not obviously better than those from existing approaches.

**Questions:**

- The authors claim that they do not use native 4K data for training. Have the authors tested whether incorporating native 4K data during training can lead to better results?

---

> ### Author Response · Authors · 2025-11-28
>
> ### W1:
> *“The quantitative evaluation could be more comprehensive. Currently, the paper only includes FID and IS metrics,
> which alone aren’t sufficient to fully capture the image quality. Including additional Image Quality Assessment (IQA)
> metrics such as PSNR or CLIP-IQA would provide a more complete evaluation.”*
>
> **Response**: Thank you for the suggestion. We have conducted additional evaluation using CLIP-IQA. We did not
> include PSNR, as it is known to be less meaningful for T2I tasks due to the inherent diversity of valid outputs and
> its sensitivity to pixel-level differences.
>
> #### Table 1: CLIP-IQA comparison.
>
> | Method | CLIP-IQA ↑ |
> |---|---|
> | SANA | 0.4457 |
> | Diffusion-4K | 0.3012 |
> | UltraPixel | 0.4421 |
> | URAE | 0.3369 |
> | FLUX+SR (BSRGAN) | 0.3897 |
> | DemoFusion | 0.4392 |
> | DiffuseHigh | 0.4221 |
> | HiDiffusion | 0.4021 |
> | FreCaS | 0.2704 |
> | I-MAX | 0.4381 |
> | HiFlow | 0.4407 |
> | **Scale-DiT (Ours)** | **0.4505** |
>
> ---
>
> ### W2:
> *“In terms of visual comparisons (e.g., Fig. 3 on page 6), Scale-DiT does not show clear visual superiority, especially
> when compared against the baseline HiFlow. The generated images from Scale-DIT are good, but not obviously better
> than those from existing approaches.”*
>
> **Response**: In Figure 3 (Page 6), qualitative comparisons highlight the advantages of our method. Our approach
> produces clearer and anatomically correct hand features, preserves fine suit details, and reconstructs individual
> hair strands, while URAE and SANA degrade in quality and HiFlow introduces artifacts. Similarly, our results
> show sharper tree structures, more coherent foliage, and improved skin detail. Additional comparisons in Figures
> 10 and 11 further emphasize these differences, and quantitative results in Table 1 confirm our method’s superiority.
>
> ---
>
> ### Q1:
> “The authors claim that they do not use native 4K data for training. Have the authors tested whether incorporating
> native 4K data during training can lead to better results?”
>
> **Response**: We agree that incorporating 4K data for fine-tuning can be beneficial, as such data contain high-
> quality details absent from 1K images. Following your suggestion, we replaced the VAE of Flux with the one
> from UltraFlux (https://arxiv.org/abs/2511.18050 ), a recently released model that post-trains the VAE of Flux
> using 4K data, and we have observed significant improvements in generated details. We believe that the global
> and semantic structures of generated images are captured in the latent space, while high-resolution details are
> stored in the VAE encoder and decoder. Our work mainly contributes to 4K latent-space generation, which focuses
> on efficiently modeling long-range dependencies and global structures, while leveraging low-resolution guidance to
> reduce computational cost and memory usage without compromising image quality.

---

### Official Review · Reviewer_BCgx · 2025-11-01

**Soundness:** 3
**Presentation:** 3
**Contribution:** 2
**Rating:** 4
**Confidence:** 4

**Summary:**

This paper proposes a framework to accelerate diffusion transformers (DiTs) at high resolution scales like 2K and 4K. The key insight here is to use local window attention and low-resolution global attention together, making them handle both local details and global semantics well. For implementation, the paper introduces a Hilbert curve order, making tokens within the same local window gathered in a memory-contiguous format, which benefits efficiency. Experiments are conducted on FLUX. The results indicate that the proposed method achieves better results than existing baselines like SANA, URAE, I-Max, HiFlow, etc.

**Strengths:**

1. The Hibert order is an interesting and effective method for implementation to accelerate practical efficiency.
2. The observation that allowing tokens within a local window to attend to tokens in the adjacent windows can resolve the boundary problem is useful.
3. The writing is overall coherent and easy to follow.

**Weaknesses:**

1. One major concern is the originality and novelty of the key insights. Using local attention to accelerate high-resolution generation has been explored in [a,b,c]. There are also models combining local attention and global low-resolution attention in the released models of [a]. The difference between [a] and this paper is to use neighborhood attention or window attention. And the difference between [b,c] and this paper is whether to use global low-resolution attention. These are not substantial difference.
2. The practical gain of the proposed Hibert order attention is not studied in the experiments.


[a] CLEAR: Conv-Like Linearization Revs Pre-Trained Diffusion Transformers Up, Liu et al., NeurIPS 2025.

[b] Fast Video Generation with Sliding Tile Attention, Zhang et al., ICML 2025.

[c] Grouping First, Attending Smartly: Training-Free Acceleration for Diffusion Transformers, Ren et al., arXiv 2025.

**Questions:**

1. From my experience, it is hard for FLUX itself to generate 4K images directly from noise. Why can merely changing the attention formulation and involving low-resolution attention resolve this issue without any 4K training data? More studies would be beneficial here.
2. It seems that the method achieves sub-optimal results in 2K resolution. Is it possible to use the proposed method together with previous training-free ones like I-Max and HiFlow to obtain stronger results?

---

> ### Author Response · Authors · 2025-11-28
> **Response Regarding Weakness**
>
> ### W1:
> *“One major concern is the originality and novelty of the key insights. Using local attention to accelerate high-resolution
> generation has been explored in [a,b,c]. There are also models combining local attention and global low-resolution
> attention in the released models of (a). The difference between (a) and this paper is to use neighborhood attention or
> window attention. And the difference between [b,c] and this paper is whether to use global low-resolution attention.
> These are not substantial difference.”*
>
> **Response**: We thank BCgx for pointing out the missing references. They were all published in 2025, and we
> apologize for overlooking these very recent papers.
>
> At first glance, our work shares a certain degree of similarity with them. However, there are some fundamental
> differences. Below, we provide point-by-point comparisons:
>
> [a]:
> 1. CLEAR proposes Circular Local Neighbor Attention, but the design is not fully optimized for GPU architectures.
> As discussed in their paper, the practical acceleration achieved by CLEAR does not fully meet its theoretical
> expectations. In contrast, our local window design can better align with GPU parallelism.
> 2. As shown in Fig. 5 of [c], it exhibits artifacts similar to those in Fig. 5 of our main paper, indicating that it
> cannot guarantee long-range global consistency.
>
> [b]:
> 1. Sliding Tile Attention (STA) is primarily optimized for video setups. It is conceptually similar to the Swin
> Transformer attention discussed in [a], which inherently lacks high-rank attention (Table 1 in [a]). Consequently,
> Swin Transformer–based methods generally perform worse than Neighbor Attention.
> 2. STA works well for videos because videos exhibit strong local temporal similarity. However, when generating
> long video sequences, inconsistencies emerge. Due to this limitation, the videos they generated are mainly 5-
> second videos. By analogy, for ultra-high-resolution (UHR) image generation, such methods struggle to maintain
> long-range global consistency.
>
> [c]:
> 1. The grouping strategy proposed by [c] is essentially another smart implementation using larger local windows.
> The GRAT-B and GRAT-X attentions in [c] can be viewed as larger-window Neighbor Attention and Criss-Cross
> Attention, respectively. While using larger windows can mitigate long range global-consistency issues, it cannot
> fully resolve them.
> 2. The Criss-Cross Attention in GRAT-X maintains long-range consistency better than Neighbor Attention in
> GRAT-B, as also demonstrated in [c], but can still be biased toward horizontal and vertical directions since the
> long-range attention is calculated in these two directions only.
>
> In summary, across [a], [b], and [c], there is a clear consensus that local neighbor attention effectively preserves
> local image quality. The principal challenge, however, is maintaining long-range dependencies, and none of these
> methods achieves this satisfactorily. They either require increasingly large local windows ([a], [c]) or sacrifice some
> degree of long-range consistency ([b], [c]). In contrast, we argue that long-range consistency can be maintained
> at low resolution without compromising efficiency or high-quality image generation. Therefore, we believe our
> formulation is significant!
>
> ---
>
> ### W2:
> *“The practical gain of the proposed Hibert order attention is not studied in the experiments.”*
>
> **Response**: The Hilbert-order permutation is an implementation choice that ensures the 2D local window in
> the image corresponds to the 2D local window in the key-value feature space, enabling efficient local attention
> computation on GPUs. Without it, the tokens would scatter in the key-value feature space. Consequently, the
> memory access becomes less contiguous, leading to reduced computational efficiency and slower inference. By using
> the Hilbert-order permutation, we maintain spatial locality, which not only accelerates attention computation but
> also preserves high-quality feature interactions within each local window. Other types of ordering can also be used
> as long as they ensure efficient calculation of local window attention.

---

> > ### Author Response · Authors · 2025-11-28
> > **Response Regarding Questions**
> >
> > ### Q1:
> > *“From my experience, it is hard for FLUX itself to generate 4K images directly from noise. Why can merely changing
> > the attention formulation and involving low-resolution attention resolve this issue without any 4K training data? More
> > studies would be beneficial here.”*
> >
> > **Response**: Our method calculates attention within local windows, while global information is maintained through
> > low-resolution representations. By avoiding direct computation of global attention, our approach preserves genera-
> > tion quality even without 4K data for global fine-tuning. In contrast, other formulations, such as linear attention,
> > require global attention to be computed, making 4K data necessary.
> >
> > We agree that incorporating 4K data for fine-tuning can be beneficial, as such data contain high-quality details
> > absent from 1K images. Following your suggestion, we replaced the VAE of Flux with the one from UltraFlux
> > (https://arxiv.org/abs/2511.18050 ), a recently released model that post-trains the VAE of Flux using 4K data, and
> > we have observed significant improvements in generated details. We believe that the global and semantic structures
> > of generated images are captured in the latent space, while high-resolution details are stored in the VAE encoder
> > and decoder. Our work mainly contributes to 4K latent-space generation, which focuses on efficiently modeling
> > long-range dependencies and global structures, while leveraging low-resolution guidance to reduce computational
> > cost and memory usage without compromising image quality.
> >
> > ---
> >
> > ### Q2:
> > *“It seems that the method achieves sub-optimal results in 2K resolution. Is it possible to use the proposed method
> > together with previous training-free ones like I-Max and HiFlow to obtain stronger results?”*
> >
> > **Response**: As discussed in Q1, we believe the “sub-optimal” results may possibly stem from the VAE of Flux,
> > which was originally trained for 1K image generation. While we successfully generated the 4K latents, the sub-
> > optimal 1K decoder cannot fully reveal the potential of our work.
> >
> > I-MAX and HiFlow leverage a low-resolution denoising path to achieve ultra-high-resolution generation. Their
> > approach is orthogonal to ours, as it primarily focuses on stabilizing high-resolution results through progressive
> > resolution enhancement. In contrast, our method aims not only to generate 4K images but to do so in an efficient
> > manner. Combining the two approaches could potentially leverage the strengths of both, we consider this an
> > interesting direction for future work.

---

### Author Response · Authors · 2025-11-28
**General Response**

## General Response
We thank our ICLR reviewers for their thoughtful feedback. All reviewers found the paper’s core approach effective,
noting that combining local window attention with low-resolution global attention provides a practical and coherent
solution for ultra-high-resolution generation. BCgx additionally praised the Hilbert-curve token ordering and noted
that cross-window attention is useful for resolving boundary issues. SUzV highlighted that mixing low-resolution
and high-resolution data for training is both interesting and effective, and further praised our strong practical
advantages such as reduced memory usage, faster inference, and low-cost LoRA fine-tuning. nPgv considered our
method straightforward and novel and supported by strong experimental results.

An enhanced version of the paper revised to address reviewers’ questions has been uploaded, with modified
sections marked in blue. Below is a summary of changes:
1. The following references are added and discussed:
- (a) CLEAR: Conv-Like Linearization Revs Pre-Trained Diffusion Transformers Up, Liu et al., NeurIPS 2025.
- (b) Fast Video Generation with Sliding Tile Attention, Zhang et al., ICML 2025.
- (c) Grouping First, Attending Smartly: Training-Free Acceleration for Diffusion Transformers, Ren et al., arXiv 2025.
- (d) HiDiffusion: Unlocking High-Resolution Creativity and Efficiency in Low-Resolution Trained Diffusion Models, ECCV 2024.
- (e) FreCaS: Efficient Higher-Resolution Image Generation via Frequency-aware Cascaded Sampling, ICLR 2024.
2. Additional evaluation using CLIP-IQA, along with comparisons to FreCaS and HiDiffusion, is now included in the quantitative comparison; their corresponding visualizations appear in the qualitative comparison.
3. Figures (especially those tied to the prompts) have been revised for improved readability.
4. We added a demonstration figure in appendix illustrating the advantage of the Hilbert-order token permutation.
5. We added a discussion to evaluate the effect of fine-tuning the VAE on 4K data.

---

### Meta-Review · Area_Chair_4Jpd · 2025-12-28

**Summary:**

There are three reviews of this paper.

Reviewer BCgx’s main concerns focus on the originality and novelty of the key insights, and the practical gain of the proposed Hibert order attention.

Reviewer SUzV’s main concerns focus on the insufficient quantitative evaluation and the unclear visual superiority.

Reviewer nPgv’s concerns focus on the organization and presentation, the missing related work, and the lack of training settings.

**Reviewer Concerns:**

Reviewer BCgx did not provide feedback on the authors’ rebuttal. The AC feels that part of his/her concerns are addressed, but the key concern about the novelty of the proposed method is not fully addressed because the authors admitted that they overlooked some of the similar works published in 2025. Reviewers SUzV and nPgv did not provide feedback on the authors’ rebuttal, but the AC feels that their concerns are addressed.

**Reviewer Scores:**

The ACs think that Reviewer BCgx will keep his/her rating of 4 since some key concerns are not addressed, and Reviewers SUzV and nPgv will keep their ratings of 6. The final scores are likely to be 4, 6, 6. Unfortunately, this paper cannot be accepted due to the competitiveness of ICLR.

---

### Decision · Program_Chairs · 2026-01-26

Reject